# Recognition and differentiation of dural puncture click sensation: A subjective and objective prospective study of dural puncture forces using fine-gauge spinal needles

Isao Utsumi[ID]*[☯], Tomasz Hascilowicz[☯], Sachiko Omi

Department of Anesthesiology, The Jikei University School of Medicine, Tokyo, Japan

☯ These authors contributed equally to this work.
* isa049@jikei.ac.jp

**Data Availability Statement:** The data underlying this study are publicly available at https://doi.org/10.5061/dryad.zs7h44j95.

## Abstract

### Background

We hypothesized that the click perceived when puncturing the dura-arachnoid with fine-gauge spinal needles can be subjectively identified, and investigated whether it may be distinguishable among different needle types.

### Methods

Subjective and objective evaluations were performed. First, physicians punctured the polyamide film or porcine dura mater (n = 70 and n = 20, respectively) with seven types of spinal needles and numerically evaluated the perceived click sensations. Using an 11-point numerical rating scale (from "0" for "no click sensation" to "10" for "the strongest click perceived") data, subjective differentiation among needle types was assessed. Second, in the objective part of the study, total forces elicited by polyamide film or porcine dura mater punctures with each needle were measured using a biomechanical testing device, and load-displacement curves evaluated. Third, the results of subjective and objective evaluations were compared.

### Results

All participants recognized the click and could discriminate among needles of different tip shape. The load-displacement curves for polyamide film and porcine dura mater were similar and needle-specific. The subjective numerical rating scale values corresponded well with the objectively measured changes in total forces ($R^2$ = 0.862 and $R^2$ = 0.881 for polyamide film and porcine dura mater, respectively), indicating that an increase in the largest drop in total force value of 0.30 N or 0.21 N would produce an increase of numerical rating scale value of 1 for polyamide film and porcine dura mater, respectively.

**Funding:** The authors declare that no funding have been received for the study.

**Competing interests:** The authors have declared that no competing interests exist.

## Conclusions

We provide an objective proof of the click sensation felt upon dural puncture using different fine-gauge spinal needles. Click recognition could be used as an additional indicator of successful spinal puncture.

## Introduction

Dura-arachnoid mater (hereafter referred to as dura mater) puncture during spinal anesthesia involves blindly advancing a needle until it reaches the subarachnoid space when the return of cerebrospinal fluid (CSF) indicates correct needle placement. This technique relies on skill and experience, as it is based solely on sensation perceived by the practitioner's fingertips. A "click" sensation or "loss of resistance" after the dura has been breached has been considered as an indicator of dural puncture allowing anesthesiologists to stop needle advancement before it impinges upon spinal nerves [1–3].

Existence of the click sensation was first reported in 1951 by Whitacre, who introduced the pencil-point spinal needle [4]. Subsequently, the click was recognized as *dural* puncture resistance, and, since the 1990s, there have been reports on click recognition rates in relation to dural puncture resistances produced by different spinal needles [5–7]. Existence of the click has been recognized and is referred to as the "dural puncture click", but detailed studies clarifying from which tissue the click originates have not been performed. Since Whitacre's report, spinal needles have undergone structural improvements, and fine-gauge pencil-point needles (25G or thinner) were developed and are currently recommended to reduce the risk of post-spinal headache (PSH) [8–11]. However, due to their thin structure, length, and bending during passage through skin, subcutaneous tissues, muscles, and interspinous and flavum ligaments before reaching the dura mater, they have been more difficult to use. To overcome these technical difficulties and prevent tissue particles from being carried into the subarachnoid space ("coring") [12], spinal needle introducers have been developed (double-needle technique) [13]. In addition, it has been observed that click recognition is less apparent with fine-gauge, than with large-gauge needles [4, 5, 14], and the incremental needle advancement technique has been recommended [1]. With this technique, however, the needle may be advanced too far anteriorly in the neuraxial canal, touch a spinal nerve root, or even exit the dura sac on its anterior aspect without noticeable CSF return [15–18]. The design of fine-gauge spinal needles also eliciting a well-perceived click sensation might be advantageous to lower the risk of PSH and increase accuracy of needle advancement.

Experiments with fresh human dura mater are difficult to perform due to several limitations: the dura is easily breakable during separation from the spinal cord, it has uneven fibrous structure (thinner in the cervical and thicker in the lumbar region), it cannot be collected in large quantities and is difficult to preserve. Needle manufacturers use synthetic artificial dura mater, but its reliability as a substitute for fresh human dura during punctures with spinal needles has not been investigated.

We hypothesized that fine-gauge spinal needles would produce subjectively perceivable click sensations upon puncture of synthetic dura or fresh porcine dura mater, and that these click sensations would be distinguishable among various spinal needle types, and would reflect objectively measured forces elicited by punctures of either synthetic dura and fresh porcine dura mater. If our hypothesis was true, it would prove that the click sensation is produced, at least in part, by puncture of the isolated dura mater. Furthermore, we aimed to examine whether the artificial dura demonstrates similar characteristics to fresh porcine dura mater.

## Materials and methods

The study was designed as an observational study and conducted following approval from the Institutional Ethics Committee of The Jikei University School of Medicine for Biomedical Research [Approval No. 31-276(9775) Date. 09/12/2019].

The study included subjective and objective evaluations and in both, a synthetic membrane and porcine dura mater (PDM) were used. Subjective evaluations assessed: (S) whether or not practitioners can recognize the click upon puncture of either synthetic membrane or PDM with various fine-gauge spinal needles and whether or not they can differentiate among spinal needles with different tip shapes, gauges, and manufacturers. Objective evaluations included: (O) measurements of forces produced during mechanical punctures of either synthetic or PDM with spinal needles of different types, and comparisons of force characteristics between synthetic and PDM; and (S-O) evaluation of the relationship between subjective and objective findings.

Seven types of fine-gauge spinal needles: 27G cutting needle, Unisis® (Tokyo, Japan); 27G pencil-point needle, Unisis® (Tokyo, Japan); 25G cutting needle, Unisis® (Tokyo, Japan); 25G pencil-point needle, Unisis® (Tokyo, Japan); 25G cutting needle, B.Braun® (Tokyo, Japan); 25G pencil-point needle, B.Braun® (Tokyo, Japan); 25G open-end needle (with a cutting tip), Dr. Japan® (Tokyo, Japan) available in Japan were selected for comparisons. In both subjective and objective experiments, all needles were advanced through introducers included in the needle kits, and were used with stylets in place. A polyamide (PA) film (PAF) (Portex® Nylon Film C-gauge) of 50 μm thickness was used as substitute material for the artificial dura mater. Fresh PDM was collected as a by-product from a meat-processing plant (Tokyo Shibaura Zoki Co., Ltd.; Tokyo, Japan), and excised from the spinal cord within 1 h after the pigs (Duroc, Landrace, and Large White Yorkshire breeds, approximately 6 months old) were sacrificed. Samples of fresh PDM cut lengthwise at 4-cm intervals were stored in the physiological saline solution at 2–4°C until immediately before experimental use. All samples were used within 24 h of preparation, and the thickness of each excised dura sample measured using a micrometer. The equipment used in this study was loaned by the Unisis Corporation (Tokyo, Japan).

### Subjective experiments (S)

All participants in the subjective portion of the study were physicians with clinical experience in spinal anesthesia. They were given oral explanations about the study nature and the use of collected data and gave written consent for participation. Data were collected between April 2018 and November 2020.

The seven spinal needles were inserted through a 20-G introducer needle fixed in a hard sponge block. The needles were set-up to enable the perpendicular puncture of either the synthetic or porcine dura (Fig 1A). Both dura were fixed in position by rubber sheets with a middle excised portion.

The number of participants was 70 in the PAF group and 20 in the PDM group due to limited availability of the porcine dura. For subjective click recognition, participants punctured the synthetic or porcine dura with the seven spinal needles, and evaluated the click sensations using an 11-point numerical rating scale (NRS) with 0 meaning that the click is not recognized and 10 standing for the strongest click perceived; no negative values were allowed. A reference score of "10" was assigned *a priori* for the Dr. Japan® 25-G open-end needle based on results of preliminary evaluations, in which five senior anesthesiologists (> 10 years of practice) all perceived the strongest click with this needle compared to the other six.

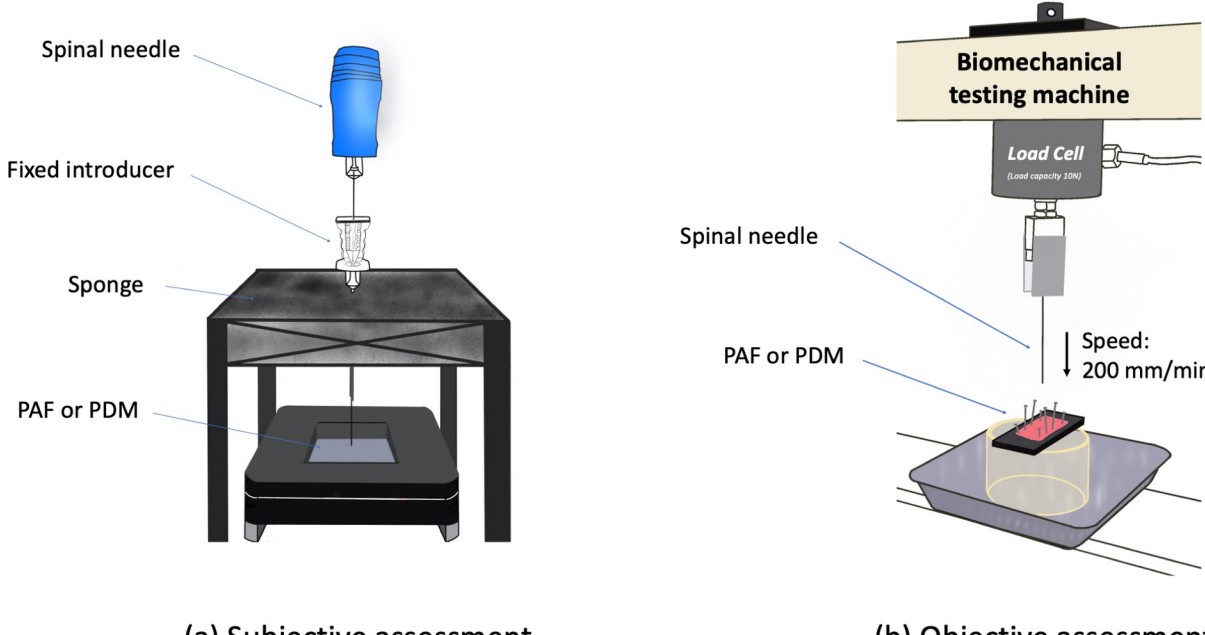

**Fig 1. Experimental models of dural puncture with various spinal needles.** (a) Subjective assessment. Participants were asked to randomly puncture the artificial dura mater (PAF) or porcine dura mater (PDM) at different locations using different spinal needles with the stylet in place thorough the needle introducer. The needle hubs were covered with a colored tape to make their original colors and shapes indistinguishable. (b) Objective measurements of puncture resistance forces using the biomechanical testing machine (AG-I; Shimazu Corporation, Kyoto, Japan). The device was equipped with a 10-N load cell unit connected to an analog measurement circuit, and the obtained data were processed through an analog filter before A/D conversion. Samples (4 × 4 cm in size) of either 50 µm PAF or of PDM were mounted with pins on the testing device. The different types of spinal needles were fixed after passing through the 20 G introducer needle and loaded so that the dura mater could be punctured perpendicularly at a constant velocity of 200 mm min$^{-1}$. Abbreviations: PAF–PA film, PDM–porcine dura mater.

In the single-blind experimental settings, participants were first instructed on the defined NRS range. Next, they were asked to puncture the dura with the Dr. Japan® needle, with an assigned score of 10, and then to perform punctures with the remaining six needles in a random order. The needle types were not disclosed and the participants, blinded to the six needles, assigned the NRS scores according to their subjective click sensations. After each set of seven punctures, when a participant needed an additional puncture to assign the final NRS score, the dura was slightly moved (approximately 5–10 mm) to ensure that each puncture was performed in a clean part of the dura. Punctures with each spinal needle could be repeated as many times as required; however, to avoid potential "material fatigue" and attachment of film particles to the needle, the number of punctures with a single needle was limited to 10. Based on the obtained NRS values, comparisons between spinal needles with different tip shapes, thickness, and manufacturers for both PAF and PDM were performed.

## Objective experiments (O)

Dural puncture force was measured using a biomechanical testing device (AG-I; Shimazu Corporation, Kyoto, Japan). Samples (4 cm × 4 cm in size) of either the 50 µm-PA film or PD were mounted with pins onto the testing device. The spinal needles were loaded so that the needle punctured the dura at 90˚ at a constant velocity of 200 mm min$^{-1}$ through introducers (Fig 1B). Load-displacement curves with needle penetration depth [mm] on the x-axis and force [N] on the y-axis were recorded for each needle.

In the preliminary studies, the results obtained for PAF were more consistent (smaller ± SD from the mean) for all needles than those of PDM (larger ± SD from the mean); therefore, in the final bench experiment, the number of PAF punctures was set to 10 per needle type and the number of PDM punctures set to 20 per needle type. In PDM experiments, each needle was not used >10 times due to an observed tendency towards increased measured forces when the same needle was used ≥20 times. The puncture loci on either PAF or PDM were randomly selected.

Changes in measured force produced by either PAF and PDM were recorded and represented as load-displacement curves for each needle. These curves were compared with regard to their shapes, peak force values required for dural puncture, plateau values, and displacement depths (calculated from the time of puncture). Curve shape similarities were analyzed with the Euclidean distance calculation algorithm. First, for each spinal needle the mean values of each point on the load-displacement curve were separately calculated for PAF and PDM data, respectively, and within the time interval 0 to 1.19 sec. This time interval was selected because 1.20 sec from the start of data collection, all needles had already penetrated both the PAF and PDM and no large changes in curve shapes had been observed after that time point. Since the characteristic features of each load-displacement curve (needle-membrane interaction "events") occurred at different time points for each needle and membrane, the Euclidean distance calculations were performed to adjust these time points and compare the curve data (curve shapes). The closest distances between the respective PAF and PDM curves for each needle, that is, the Euclidean distances, were calculated for different combinations of the mean PAF data for a particular needle (reference curve data) with those of the PDM curve data for that needle at predetermined intervals (0–30 steps backwards on the time axis). The closest distances obtained between the PAF and PDM load-displacement curve data out of performed calculations for each needle were then used for comparisons between the PAF curve data (reference curve) and the PDM curve data of all needles in a similar fashion. If, for a given needle, the closest obtained Euclidean distance between the PAF (reference curve) and PDM data was smaller than those between the PAF (reference curve) data and other PDM curve data, the similarity between curves was considered the closest. Even if the Euclidean distance values were numerically smaller in other needle-PAF-PDM combinations, the closest similarity was only assigned within the data set for the combination of a given needle, its PAF data and PDM data of all needles.

The relationship between subjective and objective (S-O) findings was analyzed by comparing the mean NRS values with the largest drop in total force (LDTF) mean values on the load-displacement curve for each needle and dura using linear regression analysis. The value of LDTF between the local maximum and minimum on the load-displacement curve was assumed to represent the perceived click sensation better than other minor local force differences due to the very short stimulus duration. The LDTF value for each needle and dura was calculated as the difference between the mean value of the local maximum and minimum that would produce the largest difference in the respective data set (S1 Fig).

## Statistical analysis

The required sample sizes for subjective and objective experiments were not calculated because both were bench experiments. Statistical data comparisons were performed using the statistical software IBM SPSS Statistics version 24.0 for Windows (IBM Corporation, Chicago, IL, U.S. A.). Click sensations among the different needle types were compared using the Kruskal-Wallis H test, and post hoc Dunn's procedure with Bonferroni correction were used to perform pairwise comparisons between needle types, separately for PAF and PDM; significant

probabilities were adjusted and data were assumed to be nonparametric. Linear regression analysis was used to assess the relationship between the NRS and LDTF values for each needle, separately for PAF and PDM. The significance level for statistical tests was set at 0.05. All data are expressed as mean and standard deviation. Load-displacement curve similarity comparisons for each needle and dura were conducted in R ver. 4.0.3 with Euclidean distance calculations codes included in the TSdist package [19].

## Results

(S) Both anesthesiologists and surgeons participated in the study (in Japan surgeons regularly perform spinal anesthesia). In the PAF group, 70 physicians experienced in spinal anesthesia were enrolled. Among them, there were 40 anesthesiologists, 30 surgeons; 51 men and 19 women; four of them had 3–5 years' experience, 48 had 5–10 years, and 18 had 10 years or more experience in spinal anesthesia. There were 20 physicians experienced in spinal anesthesia in the PDM group: seven anesthesiologists, 13 surgeons; 14 men and six women; nine of them had 3–5 years' experience, five had 5–10 years, and six had 10 years or more experience in spinal anesthesia.

No data were missing. Though all 70 participants in the PAF group were instructed that the click perceived with the Dr. Japan® 25 G open-end needle used as a reference had the score of 10, on the final comparison of click sensations, 67 participants (97%) felt the strongest click with this needle compared to other needles. The remaining three participants assigned an NRS score of 11 to the Unisis® 25 G pencil-point needle, and two assigned a score of 12 to the B. Braun® 25 G pencil-point needle (Table 1). Among all 70 participants in the PAF group, two (2.8%) reported not perceiving the click with the 27 G cutting needle, resulting in a score of 0 (Table 1). Scores >10 or equal to 0 were not assigned by participants in the PDM group (Table 2).

Pairwise comparisons between the NRS values assigned by each participant for each needle type and obtained for either PAF or PDM were performed to examine the possibility of discrimination between needle tip shape, thickness, and manufacturer. The results of these analyses are shown in Table 3. In general, the differences between cutting and pencil-point shape were well recognized in both PAF and PDM groups (with the exception of the B. Braun® 25 G needles that were less distinguishable during PDM punctures). After adjustment for other needle characteristics (needle tip shape, thickness, or manufacturer), recognizable differences between needle thickness or manufacturer were not significant (Table 3).

**Table 1. PA film (n = 70).** Numerical rating scores of click sensations perceived with different spinal needles.

| Spinal needle | | | | Numerical Rating Scores—click sensations - | | | | | | | | | | | | | | |
|---|---|---|---|---|---|---|---|---|---|---|---|---|---|---|---|---|---|---|
| Tip shape | Gauge | Manufacturer | n | 0 | 1 | 2 | 3 | 4 | 5 | 6 | 7 | 8 | 9 | 10 | 11 | 12 | Mean | SD | Median |
| Cutting | 27 G | Unisis® | 70 | 2 | 17 | 28 | 11 | 9 | 0 | 2 | 0 | 1 | 0 | 0 | 0 | 0 | 2.31 | 1.4 | 2 |
| Cutting | 25 G | Unisis® | 70 | 0 | 4 | 14 | 19 | 13 | 9 | 8 | 1 | 2 | 0 | 0 | 0 | 0 | 3.67 | 1.6 | 3 |
| Cutting | 25 G | B-Braun® | 70 | 0 | 6 | 7 | 12 | 14 | 10 | 6 | 7 | 5 | 3 | 0 | 0 | 0 | 4.49 | 2.2 | 4 |
| Pencil-point | 27 G | Unisis® | 70 | 0 | 1 | 3 | 3 | 18 | 17 | 14 | 10 | 4 | 0 | 0 | 0 | 0 | 5.13 | 1.5 | 5 |
| Pencil-point | 25 G | Unisis® | 70 | 0 | 0 | 3 | 3 | 5 | 18 | 13 | 17 | 8 | 2 | 0 | 1 | 0 | 5.93 | 1.7 | 6 |
| Pencil-point | 25 G | B-Braun® | 70 | 0 | 0 | 1 | 1 | 5 | 4 | 8 | 12 | 20 | 14 | 3 | 0 | 2 | 7.39 | 1.9 | 8 |
| Open-end | 25 G | Dr. Japan® | 70 | 0 | 0 | 0 | 0 | 0 | 0 | 0 | 0 | 0 | 0 | 70 | 0 | 0 | 10 | 0 | 10 |

The Dr. Japan needle was assigned a reference of 10 *a priori*.

Participants (n = 70): Anesthesiologists:Surgeons = 40:30; Men:Women = 51:19; Experience in spinal anesthesia: 3–5 years (n = 4), 5–10 years (n = 48), ≥10 years (n = 18).

**Table 2. Porcine-dura mater (n = 20).** Numerical rating scores of click sensations perceived with different spinal needles.

| Spinal needle | | | | Numerical Rating Scores—click sensations - | | | | | | | | | | | | | | | |
| Tip shape | Gauge | Manufacturer | n | 0 | 1 | 2 | 3 | 4 | 5 | 6 | 7 | 8 | 9 | 10 | 11 | 12 | Mean | SD | Median |
|---|---|---|---|---|---|---|---|---|---|---|---|---|---|---|---|---|---|---|---|
| Cutting | 27 G | Unisis® | 20 | 0 | 5 | 6 | 5 | 3 | 1 | 0 | 0 | 1 | 0 | 0 | 0 | 0 | 2.45 | 1.2 | 2 |
| Cutting | 25 G | Unisis® | 20 | 0 | 3 | 5 | 5 | 5 | 1 | 1 | 0 | 0 | 0 | 0 | 0 | 0 | 2.95 | 1.4 | 3 |
| Cutting | 25 G | B-Braun® | 20 | 0 | 0 | 1 | 5 | 2 | 9 | 1 | 2 | 0 | 0 | 0 | 0 | 0 | 4.5 | 1.4 | 5 |
| Pencil-point | 27 G | Unisis® | 20 | 0 | 0 | 2 | 2 | 2 | 4 | 2 | 0 | 8 | 0 | 0 | 0 | 0 | 5.7 | 2.2 | 5.5 |
| Pencil-point | 25 G | Unisis® | 20 | 0 | 0 | 0 | 0 | 3 | 0 | 12 | 3 | 1 | 1 | 0 | 0 | 0 | 6.1 | 1.2 | 6 |
| Pencil-point | 25 G | B-Braun® | 20 | 0 | 0 | 0 | 0 | 0 | 2 | 0 | 10 | 1 | 7 | 0 | 0 | 2 | 7.55 | 1.3 | 7 |
| Open-end | 25 G | Dr. Japan® | 20 | 0 | 0 | 0 | 0 | 0 | 0 | 0 | 0 | 0 | 0 | 70 | 0 | 0 | 10 | 0 | 10 |

The Dr. Japan needle was assigned a reference of 10 *a priori*.

Participants (n = 20): Anesthesiologists: Surgeons = 7:13; Men: Women = 14:6; Experience in spinal anesthesia: 3–5 years (n = 9), 5–10 years (n = 5), ≥10 years (n = 6).

Kruskal-Wallis tests were performed separately for each spinal needle and for PAF and PDM with the mean NRS values. The distributions of NRS values assigned for each needle (n = 70 and n = 20, for PAF and PDM, respectively) were not similar and significantly different among needles, $\chi^2(6) = 320.903$, $p < 0.001$ and $\chi^2(6) = 105.108$, $p < 0.001$ for PAF and PDM, respectively. Adjusted p values in the Table represent results of pairwise comparisons among the needles using the post hoc Dunn's method with a Bonferroni correction, separately for PAF and PDM. Statistically significant differences are marked with an asterisk.

(O) The load-displacement curves obtained during mechanical punctures of the PA film and the porcine dura mater are shown in Fig 2.

All needles produced load-displacement curves with two peaks. This pattern was preserved for both PAF and PDM but was characteristically different between cutting and pencil-point needles. On all curves, the force values did not return to the initial value (0 N) but remained at a plateau value different for each needle. The Dr. Japan® 25 G open-end needle had only one marked peak of force values, which, compared to other needles, was also the highest for both PAF and PDM (Fig 2).

Load-displacement curve shapes of synthetic and porcine dura mater revealed visual similarities between the corresponding curve shapes produced by each spinal needle (Fig 2). These shape similarities were analyzed by calculation of the Euclidean distances at preset intervals (0–30 steps on the time axis) between mean force values obtained for each needle with PAF with those of the PDM for the same type of needle versus those of other needle. For each needle, the number of steps when the distances between PAF and PDM curves were the smallest was different and varied from 7 to 19 steps. For each needle, the closest distances obtained by

**Table 3. Comparison of click sensations (NRS values) among needles.**

| Needle characteristics | | | vs | | p value | |
|---|---|---|---|---|---|---|
| | | | | | PAF | PDM |
| Tip shape | Unisis® (27 G) | cutting | vs | pencil | 0* | 0.002* |
| | Unisis® (25 G) | cutting | vs | pencil | 0* | 0.003* |
| | B. Braun® (25 G) | cutting | vs | pencil | 0* | 0.009* |
| Thickness | cutting needle (Unisis®) | 27 G | vs | 25 G | 0.058 | 1 |
| | pencil-point needle (Unisis®) | 27 G | vs | 25 G | 1 | 1 |
| Manufacturer | 25 G cutting needle | Unisis® | vs | B. Braun® | 1 | 1 |
| | 25 G pencil-point needle | Unisis® | vs | B. Braun® | 0.121 | 1 |

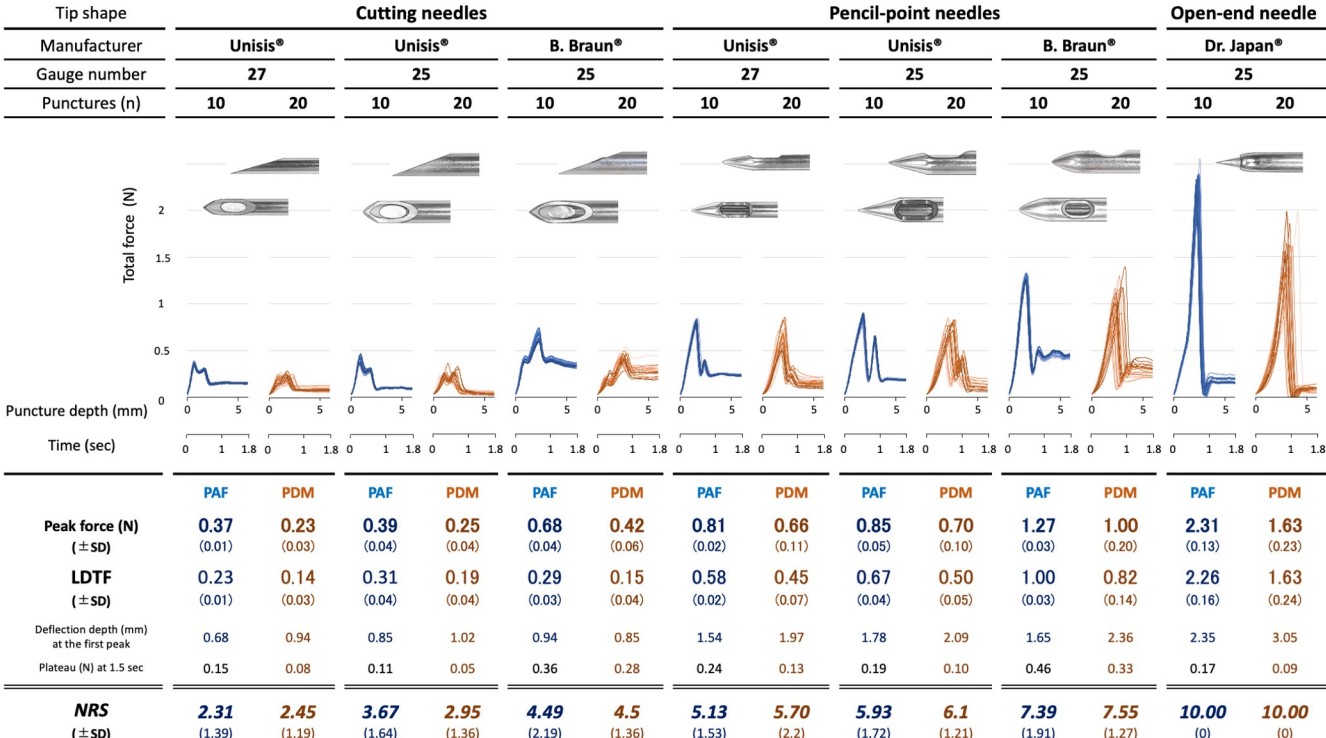

| Tip shape | Cutting needles | | | | | | Pencil-point needles | | | | | | Open-end needle | |
|---|---|---|---|---|---|---|---|---|---|---|---|---|---|---|
| Manufacturer | Unisis® | | Unisis® | | B. Braun® | | Unisis® | | Unisis® | | B. Braun® | | Dr. Japan® | |
| Gauge number | 27 | | 25 | | 25 | | 27 | | 25 | | 25 | | 25 | |
| Punctures (n) | 10 | 20 | 10 | 20 | 10 | 20 | 10 | 20 | 10 | 20 | 10 | 20 | 10 | 20 |
| | PAF | PDM | PAF | PDM | PAF | PDM | PAF | PDM | PAF | PDM | PAF | PDM | PAF | PDM |
| Peak force (N) (±SD) | 0.37 (0.01) | 0.23 (0.03) | 0.39 (0.04) | 0.25 (0.04) | 0.68 (0.04) | 0.42 (0.06) | 0.81 (0.02) | 0.66 (0.11) | 0.85 (0.05) | 0.70 (0.10) | 1.27 (0.03) | 1.00 (0.20) | 2.31 (0.13) | 1.63 (0.23) |
| LDTF (±SD) | 0.23 (0.01) | 0.14 (0.03) | 0.31 (0.04) | 0.19 (0.04) | 0.29 (0.03) | 0.15 (0.04) | 0.58 (0.02) | 0.45 (0.07) | 0.67 (0.04) | 0.50 (0.05) | 1.00 (0.03) | 0.82 (0.14) | 2.26 (0.16) | 1.63 (0.24) |
| Deflection depth (mm) at the first peak | 0.68 | 0.94 | 0.85 | 1.02 | 0.94 | 0.85 | 1.54 | 1.97 | 1.78 | 2.09 | 1.65 | 2.36 | 2.35 | 3.05 |
| Plateau (N) at 1.5 sec | 0.15 | 0.08 | 0.11 | 0.05 | 0.36 | 0.28 | 0.24 | 0.13 | 0.19 | 0.10 | 0.46 | 0.33 | 0.17 | 0.09 |
| NRS (±SD) | 2.31 (1.39) | 2.45 (1.19) | 3.67 (1.64) | 2.95 (1.36) | 4.49 (2.19) | 4.5 (1.36) | 5.13 (1.53) | 5.70 (2.2) | 5.93 (1.72) | 6.1 (1.21) | 7.39 (1.91) | 7.55 (1.27) | 10.00 (0) | 10.00 (0) |

**Fig 2. Load-displacement curves obtained by punctures of PAF and PDM with different spinal needles.** Values are represented as mean ± SD. Colors represent curves / values for: PAF–blue and PDM–dark orange. Abbreviations: LDTF–largest drop in total force, PAF–PA film, PDM–porcine dura mater.

the Euclidean distance calculations were those between PAF and PDM curve data of the same needle; these distances were larger when PDM curve data of other needles were compared with the PAF (reference curve) data. This indicated that within the given set of needle-PAF-PDM data, the curve shapes were the closest when PAF and PDM were punctured with the same needle, and indirectly suggested that PAF and PDM elicit similar forces when punctured with the same needle. The PDM curves demonstrated a right shift on the x-axis compared to PAF curves (Figs 2, 3) for the respective needle. Differences in values of the mean peak force, plateau, and depth of displacement for PAF and PDM are shown in Fig 2. Mean peak forces and plateaus tended to be lower for each needle puncturing PDM compared to PAF but mean depths of displacement were higher for PDM than PAF.

(S-O) The assigned mean LDTF values on the load-displacement curves obtained by PAF punctures were two- to five-fold lower for cutting needles compared to those of the pencil-point needles. The largest LDTF on the load-displacement curve was observed with the Dr. Japan® 25 G open-end needle, almost 10-times larger than the difference of the Unisis® 27 G cutting needle (Fig 2).

Among all fine spinal needles and data items obtained from the objective part of the study for each needle, the increasing order of LDTF values corresponded well to the order of increasing mean NRS values subjectively assigned for the respective needles; that is, the lowest LDTF observed with the Unisis® 27 G cutting needle matched to the lowest mean NRS value, and the highest LDTF observed with the B. Braun® 25 G pencil-point needle matched to the highest mean NRS score for the pencil-point needles, and so on. For both PAF and PDM, mean LDTF values significantly predicted NRS values [$F_{(1,5)}$ = 38.536, p = 0.002, adjusted $R^2$ = 0.862 for PAF, and $F_{(1,5)}$ = 45.477, p = 0.001, adjusted $R^2$ = 0.881 for PDM, respectively]

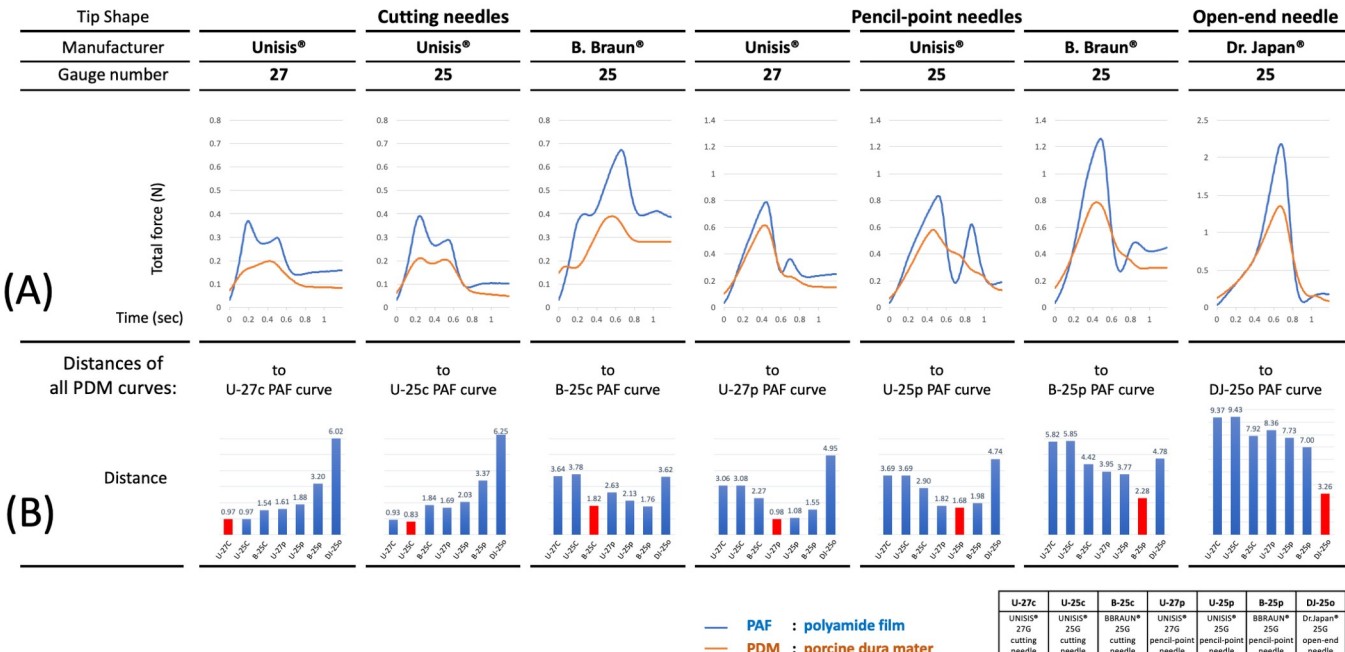

**Fig 3. Similarity of load-displacement curves analyzed with the Euclidean distance calculation algorithm.** (A) For each spinal needle, the mean values for each point on the load-displacement curve of all data sets were separately calculated for PAF and PDM data, and for the time interval 0 to 1.19 sec. The smallest Euclidean distance between each point on the PAF load-displacement curve to the PDM curve was then assessed by calculating Euclidean distances at set intervals (0–30 steps backwards) on the time axis. (B) The smallest distances (reflecting closest similarity between the PAF and PDM curves) for a given needle were compared to the distances obtained from calculations of the smallest distances between the PAF curve of that needle and those on the PDM curves of other needles.

indicating that an increase in LDTF value of 0.30 N or 0.21 N would produce an increase of NRS value of 1 for PAF and PDM, respectively.

## Discussion

Click sensation, previously recognized by Whitacre [4] and others [5–7], has been commonly associated with puncture of the dura mater and, accordingly, referred to as the "dural puncture click" or "loss of resistance" sensation upon puncture of the dura [1–3]. This observation was plausible since the click was usually experienced at the final stages of needle progression just before confirming the cerebrospinal fluid reflux, and it was also sensed with sleeved spinal needles advanced through heel lumens of Tuohy needles used for combined spinal epidural anesthesia [20]. Though associated with the dura mater puncture, the click might have originated from other tissues penetrated by the spinal needle (e.g., skin, subcutaneous tissues, muscles, interspinous and flavum ligaments) before reaching the dura. Thus, the "loss of resistance" sensation felt after puncture of the ligamentum flavum might have been recognized as the click [12]. However, this "loss of resistance" observed with epidural needles has not been uniformly experienced or referred to as a "click" *per se*, even though Tuohy needles have much larger diameters. The fact that the "click" has not been commonly recognized with Tuohy needles might be related to differences in the elasticity of tissues they penetrate as compared to that of the dura mater.

The prevailing consensus that a puncture of the dura mater produces the click sensation has been therefore not based on evidence. In this regard, 1) the click has been experienced as a subjective sensation during puncture of all tissue layers penetrated before the needle reaches

the subarachnoid space; 2) objective measurements of forces produced solely by puncture of the dura mater have not been performed; and 3) the exact clinical settings with patients receiving spinal anesthesia have been impossible to reproduce experimentally. We hypothesized that if puncture of the dura mater alone produces the click sensation, the click may be legitimately referred to as "dural puncture click", even if it only partially contributes to the overall click sensation in the clinical settings. We thought that this could only be examined with an isolated dura mater (fresh porcine dura mater in our study) and with experimental settings similar to those found in clinical situations when the influence of other tissues has been eliminated; that is, when fine-gauge spinal needles are advanced through needle introducers.

The results of subjective experiments clearly demonstrated that even with fine-gauge spinal needles (25 G, 27 G) the dural puncture click was perceived by almost all participants; only two out of 70 (2.8%) could not recognize it with the Unisis[®] 27 G cutting needle, the finest needle used in the PAF group. Furthermore, differences in needle tip shape were easily distinguishable among all spinal needles used in our study when both PAF and PDM were punctured. Thus, our results confirm that the click could serve as an additional tactile indicator for an appropriately performed spinal puncture.

The load-displacement curves obtained for both PAF and PDM were needle-type-specific: closer similarities were observed between PAF and PDM curves obtained for the same needle than for other needles (Fig 3).

At the defined point on the load-displacement curve, the total force elicited on the needle could be expressed as:

$$F_{total} = F_{elastic} + F_{resistive}$$

Where

$$F_{elastic} = kx \quad \text{and} \quad F_{resistive} = \mu v$$

$k$ is the stiffness coefficient of the membrane, $x$ represents deflection of the membrane, $\mu$ is the dynamic viscosity the membrane and $v$ is a relative velocity of the needle to the membrane. Before puncture (Fig 4A), the force acting on the needle is due to the elastic deformation of the

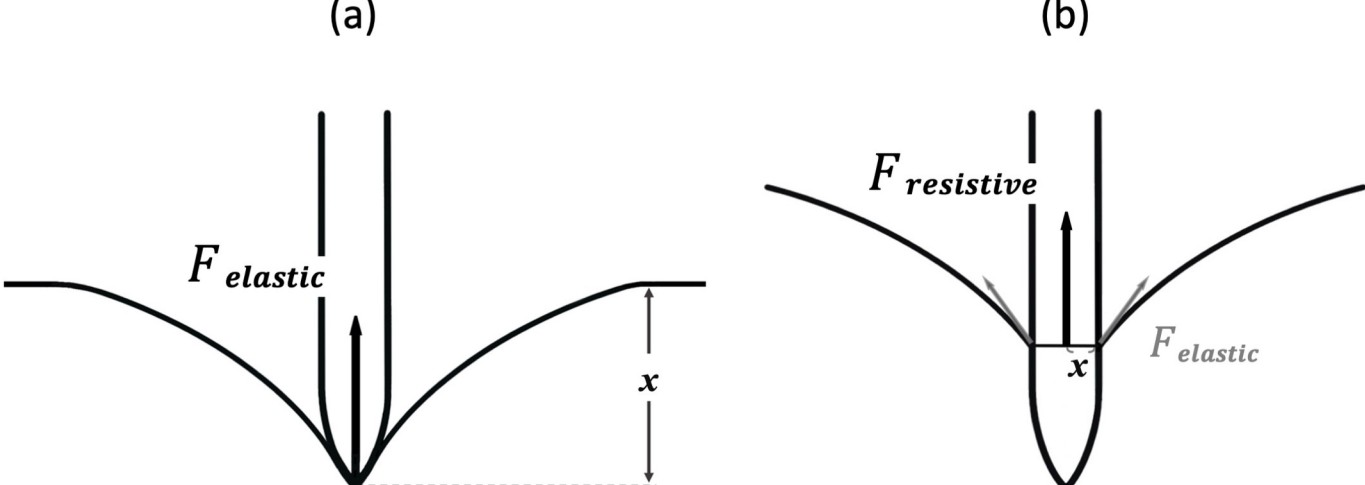

**Fig 4. Forces elicited on the spinal needle upon puncture of the dura mater.** (a) Before puncture. The force acting on the needle is due to the elastic deformation of the membrane which is represented as $F_{elastic}$. (b) After the puncture, $F_{resistive}$ is added to represent the forces produced by membrane viscosity, similar to frictional forces.

membrane which is represented as $F_{elastic}$. After the puncture (Fig 4B), $F_{resistive}$ is added to represent the forces produced by membrane viscosity, similar to frictional forces. This $F_{resistive}$ can be observed when force converges to the plateau where the effect of $F_{elastic}$ becomes minimal.

It could be assumed that larger forces should be observed for needles with a larger increase in diameter (pencil-point and open-end needles) as a rapid increase in diameter causes larger membrane deflection at a given time, resulting in larger elastic force. The elastic forces were smaller with cutting needles because the increase in needle diameter is more linear, as reflected in the direct measurements of total forces (Fig 2): all cutting needles produced lower maxima (ranges, 0.37–0.68 N and 0.23–0.42 N, for PAF and PDM, respectively) than pencil-point needles (range, 0.81–1.27 N and 0.66–1.0 N, for PAF and PDM, respectively); pencil-point needles and open-end needles having larger increases in diameter and larger cross-cut areas than cutting needles, which required larger forces to puncture the dura. The Dr. Japan® open-end needle had the largest increase in diameter amongst all and thus produced the largest force (almost twice larger than the B. Braun® 25G needle). Following dural puncture, the plateaus of needles with the same thickness were similar except for the B. Braun® 27G and 25G, which had more coarse surfaces (higher kinematic viscosity) than Unisis® and Dr. Japan® and thus also higher post-puncture plateau values.

Repeated PAF punctures in randomly selected loci produced almost overlapping load-displacement curves specific for each spinal needle. This similarity in the PDM data was not observed even if the needle-specific load-displacement patterns for the respective needles were preserved. Structural differences between the PAF and the PDM were most likely responsible for this discrepancy. The PAF is a homogeneous synthetic membrane, while PDM is a fibrous tissue composed of different size fibers and thickness ranging from 125 μm to 485 μm. The nonhomogeneous porcine dura structure was reflected in the load-displacement curve shape discrepancies obtained after repeated punctures with respective spinal needles, even though the bevels of the cutting needles were set parallel to the dura's fiber axis [21]. The comparatively lower resistance force maxima at the PDM puncture and the deeper PDM deflection immediately before its puncture were most probably related to its more compliant properties (lower stiffness coefficient) than those of the PAF (though PAF was comparatively thinner). Notwithstanding the differences between PAF and PDM, all load-displacement curves were similar. This indicates that PAF can be used as a substitute for the natural dura mater, although the abovementioned differences in viscosity and structural homogeneity should be considered.

We assigned the maximal differences between the local maxima and minima (LDTF) on load-displacement curves as those that would represent the clicks for respective spinal needles. This arbitrary assignment could have been incorrect and might be one study limitation. We could have selected the overall maximum resistance force produced by each needle at the time of PAF puncture, but one-point measurements would possibly not represent needle-specific structural differences subjectively distinguishable. This was observed when comparisons between the Unisis® cutting 27 G and 25 G needles and between the Unisis® pencil-point 27 G and 25 G needles were performed. The differences in maximum forces for these needles were very small, but could be easily distinguished. From these observations, we considered an LDTF of 0.30 N for PAF and 0.21 N for PDM as the lowest difference in total force for ready click recognition; however, further and more detailed studies on objectively measurable parameters reflecting subjective click perception are necessary.

Three important factors have been suggested for successful spinal block and selection of an "optimal" needle: 1) reliable injection of the anesthetic into the subarachnoid space, 2) low risk of nerve injury, and 3) low risk of PSH. We think that the click sensation as an additional indicator of dural puncture may potentially meet all these requirements for a successful subarachnoid injection.

1) The opening of the spinal needle must be fully open within the subarachnoid space for injection of the anesthetic. Theoretically, this can be easily obtained with cutting needles since their bevels are in the closest distance to the needle tip; however, occasionally, and particularly with fine gauge needles, the correct depth of the needle progression (usually performed in millimeter intervals and by repeated pulling out the stylet) can be difficult to define even when CSF reflux is confirmed, when only part of the needle opening is inside the subarachnoid space [22]. Consequently, in addition to the drug entering the subarachnoid space, some of it enters subdural and/or epidural spaces; therefore, the required anesthetic effect cannot be obtained. These fine-gauge cutting needles were also the ones that produced relatively weaker click sensations in our study and the click was even unrecognized with the finest, Unisis® 27G, cutting needle. If the click acted as an indicator of the needle orifice fully open within the subarachnoid space, injection of the drug would be more accurate and the anesthetic effective. Design of a fine needle that produces an easily perceivable click and that has an orifice close to its tip would potentially solve that issue.

2) Deep penetration-related nerve/spinal cord injury may develop due to either too deep progression of the sharp cutting needles (unrecognized dural puncture) or due to dura mater concavity observed especially with blunt-ended needles (pencil-point or open-end needles). In the latter case, the puncture occurs at the peak force when the needle tip is already deep into the subarachnoid space. The needle is usually inserted even further into the subarachnoid space to "overcome" the initial "resistance." Thus, both very weak (fine cutting needles) and very strong (open-end needles) click sensations would be undesirable since they would reflect either extreme sharpness of the needle or extreme concave deflection (tenting) [12] at the dura mater puncture. Notwithstanding these observations, the relationship between the depth of concave deflection of the dura mater upon puncture and nerve/spinal injury has not been studied, and 3 mm depth observed with the Dr. Japan® needle might possibly be acceptable.

3) The spinal needle type that is currently recommended for preventing PSH is a fine, pencil-point needle since it divides rather than cuts the dura mater fibers during puncture [8–11]. Notwithstanding the recommendation, some practical problems that have been reported with this type of needle that have led to its lower preferability among practitioners include the following: (a) its fineness and flexibility result in the needle bending at the time of dural puncture [15]; (b) insufficient sharpness of its tip hinders needle penetration through tissues; and (c) the need for an introducer complicates the procedure [8, 14]. All simultaneously focus on either superiority of fine-gauge non-cutting needles in regard traumatic dura mater injury or inferiority of these needles in regard to procedural issues. The click sensation, if properly recognized, might bring a solution to these contradictory issues, i.e., the finer the needle the lower the risk of PSH but the more difficult the puncture. On one hand, it would support the use of fine pencil-point needles and on the other, indicate that too thin needles complicate the procedure of spinal puncture. Incorporating the concept of the optimal click recognition into design of a needle with the optimal gauge and shape would thus be plausible and beneficial.

Ideally, such a needle would fulfil the following requirements: 1) be a fine-gauge needle close in shape to pencil-point or open-end needles; 2) elicit a click sensation corresponding to higher than approximately 0.30 N or 0.21 N LDTF upon PAF or PDM puncture, respectively; 3) have the orifice as close to the needle tip as possible [22]; and 4) do not cause too extensive concave deformation of the dura mater immediately before dural puncture. Design of such a needle can potentially increase the success rate and improve the learning curve of the safe spinal anesthesia technique.

## Limitations

In addition to the aforementioned arbitrary selection of LDTF on load-displacement curves, which indicates the inability to precisely identify when exactly the click occurs on the load-displacement curve, our study has other limitations: (a) interference of other tissues below the introducer needle tip on click sensation was not addressed; (b) the influence of skin and subcutaneous tissues entangled on the needle [23, 24] on frictional resistance was not examined; (c) the influence of needle progression speed was not analyzed; (d) differences between porcine and human dura mater were not addressed; (e) a small number of participants for puncturing the PDM; and (f) selection of Euclidean distance analysis for shape comparison of load-displacement curves.

(a) In clinical settings, the introducer tip does not necessarily reach the epidural space and in such circumstances, the spinal needle must penetrate the remaining tissues (interspinous and flavum ligaments) before puncturing the dura. This might influence the magnitude of click sensation; in our experimental settings this influence was not analyzed. Thus, our experimental settings would have resembled those with the ideal position of the introducer needle in the epidural space or those of combined spinal epidural anesthesia performed with the sleeved needle-in-needle technique.

(b) The tissues (e.g., skin, subcutaneous fatty tissue, striated muscle, connective tissue, ligaments, epidural space, etc.) penetrated before the needle reaches the dura mater may affect the force and deflection of the dura through their entanglement around the needle and thus interfere with the overall forces and click sensation. In our in vitro settings, this influence could not be fully replicated. However, punctures with fine-gauge spinal needles are usually performed with introducers designed to avoid the entanglement of tissue particles on the needle surface and in that sense our experimental settings were close to the clinical ones.

(c) The force needed for dural puncture has been reported to decrease with increased needle insertion speed [25]. Consequently, the recognition ability of puncture resistance forces would increase with decreasing needle progression speed. This issue has not been analyzed in our study where the needle progression speed was set at 200 mm/min in the Objective evaluation part and the needle insertion speed in the subjective evaluation part of the study was not determined.

(d) Although there are similarities between the porcine and human dura mater, the characteristics of human dura mater with regard to puncture resistance were not examined. The properties potentially affecting click sensation could be sex, age, disease and physiological state. Inter- and intra-observer differences in click perception between practitioners should be also taken into consideration.

(e) The number of participants in the PDM group may have been too small due to the limited availability of porcine dura.

(f) Other methods for comparisons of curve shapes have been available but the Euclidean distance calculation method was selected because it was included in the TSdist statistical package [19] of R software. Other curve comparison methods may have yielded possibly different results and such calculations were not discussed.

Notwithstanding the study limitations, we demonstrated that click sensation could be readily recognized by all participants even with fine-gauge spinal needles upon both PAF and PDM punctures, and that it was possible to differentiate click perceptions between needles of different shapes, even if these differences were minor. The subjective click perception corresponded well with objectively measured changes in total forces elicited during PAF and PDM punctures; results further strengthen the importance of the click concept, which could be incorporated into the design of an optimal spinal needle.

## Conclusion

Using a modified experimental model with synthetic dura mater and fresh porcine dura mater, it was clearly demonstrated that click sensation can be readily recognized by all participants upon puncture of isolated synthetic dura mater or fresh porcine dura mater with fine-gauge spinal needles, and that it is possible to differentiate click perceptions between needles of different shapes even if these differences are minor. The subjective perception of the click corresponded well with objectively measured changes in dural puncture forces; these results further strengthen the importance of the click concept, which when incorporated into the design of an optimal spinal needle, can potentially increase the success rate and improve the learning curve of the safe spinal anesthesia technique.

## Supporting information

**S1 Fig. Assignment of largest drop in total force (LDTF) values; typical examples.** The largest drop in total force (LDTF) between the local maximum and minimum on the load-displacement curve for each needle was defined to reflect the subjective click perception expressed in numerical rating scale values. The local maximum was the peak total force value in the load-displacement curve data, while the local minimum value was the nearest lowest total force value observed in the data set after rapid drop(s) in force when the difference with the peak total force value was the largest.
(TIF)

## Acknowledgments

The authors would like to thank Y. Matsumoto for technical assistance with the experiments and S. Hascilowicz for his helpful suggestions on the physical aspects of membrane puncture. We also thank T. Takamiya and M. Janik for assistance in analysis of curve similarities and Unisis®, Inc. of Tokyo for lending the biomechanical testing device.

## Author Contributions

**Conceptualization:** Isao Utsumi, Sachiko Omi.

**Data curation:** Isao Utsumi, Tomasz Hascilowicz.

**Formal analysis:** Isao Utsumi, Tomasz Hascilowicz.

**Investigation:** Isao Utsumi.

**Methodology:** Isao Utsumi, Tomasz Hascilowicz.

**Project administration:** Isao Utsumi.

**Supervision:** Sachiko Omi.

**Visualization:** Isao Utsumi.

**Writing – original draft:** Isao Utsumi, Tomasz Hascilowicz.

**Writing – review & editing:** Isao Utsumi, Tomasz Hascilowicz, Sachiko Omi.

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
