## [Decision Letter · Decision Letter 0]

26 Apr 2021

PONE-D-21-03864

Recognition and Differentiation of Dural Puncture Click Sensation:

A Subjective and Objective Prospective Study of Dural Puncture Forces Using Fine-Gauge Spinal Needles

PLOS ONE

Dear Dr. Utsumi,

Thank you for submitting your manuscript to PLOS ONE. After careful consideration, we feel that it has merit but does not fully meet PLOS ONE’s publication criteria as it currently stands. Therefore, we invite you to submit a revised version of the manuscript that addresses the points raised during the review process.

As identified in the review, the authors need to include additional tissue layer to ensure a greater correlation to real world applications. If this article is to be further considered for acceptance, these additional tissue layers need to be incorporated. 

We look forward to receiving your revised manuscript.

Kind regards,

Jonathan H Sherman

Academic Editor

PLOS ONE

Journal Requirements:

"Support for a part of study was provided by departmental sources (Daisan Hospital, The Jikei University School of Medicine, Tokyo)."

3. Please upload a copy of Supporting Information S1 which you refer to in your text on page 26.

Reviewers' comments:

Reviewer's Responses to Questions

**Comments to the Author**

1. Is the manuscript technically sound, and do the data support the conclusions?

Reviewer #1: Yes

Reviewer #2: Partly

2. Has the statistical analysis been performed appropriately and rigorously? 

Reviewer #1: Yes

Reviewer #2: Yes

3. Have the authors made all data underlying the findings in their manuscript fully available?

Reviewer #1: Yes

Reviewer #2: Yes

4. Is the manuscript presented in an intelligible fashion and written in standard English?

Reviewer #1: Yes

Reviewer #2: Yes

5. Review Comments to the Author

Reviewer #1: The Authors of Recognition and Differentiation of Dural Puncture Click Sensation:

A Subjective and Objective Prospective Study of Dural Puncture Forces Using Fine-Gauge Spinal Needles

set out to demonstrate that there is a perceived sensation of a click when puncturing the dura. They show that despite the needle type or gauge, all the participating physicians are able to perceive a sensation when they enter the study dura. Additionally, they show that the polymer dura is fairly close porcine dura and therefore could possibly be used in further studies. I think the article is a good start on figuring out what is the best needle for safe dural puncture w/adequate drug administration. Also, the data showing the reliability of the polymer dura shows that it could even be used in the future as a teaching agent for new physicians to understand the sensation of the click. I also feel the authors do a good job of addressing their limitations and suggesting where future research should go.

Reviewer #2: This papers provides an interesting perspective and analysis of the force required to puncture the dura in a porcine and synthetic model. The analysis of biomedical forces with needle size and type required to puncture dura is interesting and of clinical benefit. However, the perception of click sensation in isolated dural and perceived differences based on needle characteristics distract from the paper.

Percutaneous lumbar puncture is performed by directing a spinal needle from the skin through the subcutaneous tissue planes, ligamentous structures, and bony spinal anatomy into the dura. The perception of the “click” is diminished as a function of the depth and complexity of tissue layers perpendicular to needle trajectory. Isolating the process to include only the dura and perception of the click only at the dural surface, as this paper does, results in no real world application.

The authors should focus on the biomedical forces required to puncture dura with various needle sizes and types. Click sensation could be included if it were to be studied in a real world application such as identification of perception of click from the skin surface.

6. PLOS authors have the option to publish the peer review history of their article (what does this mean?). If published, this will include your full peer review and any attached files.

Reviewer #1: No

Reviewer #2: No

---

## [Author Response · Author response to Decision Letter 0]

4 Jun 2021

Point-by-Point Response

Please note that the changes made do not influence the content, conclusions, or framework of the paper. We have not listed below all minor changes made; however, these can be viewed as “tracked changes” in the revised manuscript.

Reviewer #1:

Comment: The Authors of Recognition and Differentiation of Dural Puncture Click Sensation: A Subjective and Objective Prospective Study of Dural Puncture Forces Using Fine-Gauge Spinal Needles set out to demonstrate that there is a perceived sensation of a click when puncturing the dura. They show that despite the needle type or gauge, all the participating physicians are able to perceive a sensation when they enter the study dura. Additionally, they show that the polymer dura is fairly close porcine dura and therefore could possibly be used in further studies. I think the article is a good start on figuring out what is the best needle for safe dural puncture w/adequate drug administration. Also, the data showing the reliability of the polymer dura shows that it could even be used in the future as a teaching agent for new physicians to understand the sensation of the click. I also feel the authors do a good job of addressing their limitations and suggesting where future research should go.

Response: Thank you for your supportive comments on the manuscript. As you have stated, we attempted to establish a base for future research on the subject.

Reviewer #2 (and Academic Editor)

Comment: This paper provides an interesting perspective and analysis of the force required to puncture the dura in a porcine and synthetic model. The analysis of biomedical forces with needle size and type required to puncture dura is interesting and of clinical benefit. However, the perception of click sensation in isolated dural and perceived differences based on needle characteristics distract from the paper.

Percutaneous lumbar puncture is performed by directing a spinal needle from the skin through the subcutaneous tissue planes, ligamentous structures, and bony spinal anatomy into the dura. The perception of the “click” is diminished as a function of the depth and complexity of tissue layers perpendicular to needle trajectory. Isolating the process to include only the dura and perception of the click only at the dural surface, as this paper does, results in no real world application.

The authors should focus on the biomedical forces required to puncture dura with various needle sizes and types. Click sensation could be included if it were to be studied in a real world application such as identification of perception of click from the skin surface.

Response: Reviewer #2 has pointed out that our experimental conditions, in which either the isolated PA (artificial dura mater) or fresh porcine dura mater were punctured with a spinal needle, do not reflect the actual clinical situation in which other tissues also influence (possibly decrease) the subjective experience of the click. Therefore, it has been suggested that a model of those tissues (e.g., skin, subcutaneous tissues, muscles, ligaments) should have been added before the click itself is analyzed.

As mentioned by Reviewer# 2, we have addressed the influence of skin and tissue in the Limitations section; however, the description in the Limitations section was misleading and could have led to a misunderstanding of the study’s rationale. Thank you for raising this issue.

We have revised the Introduction, Discussion, and relevant sections within the Materials and Methods to clarify our hypothesis and experimental design. We based our studies on the assumption that it must first be elucidated if the isolated dura mater (or its substitute) actually produces the click sensation before calling it “the dural click” and before determining the influence of other tissues on the click. In other words, if the dural puncture does not produce the click, the responsible tissue needs to be identified. However, if the dural puncture does produce the click sensation, then the effect of other tissues on the click (with relation to needle characteristics) should be examined. This has been stated more clearly in the revised manuscript as follows:

Introduction 

 Page 3, Lines 58–74

“Existence of the click sensation was first reported in 1951 by Whitacre, who introduced the pencil-point spinal needle [4]. Subsequently, the click was recognized as dural puncture resistance, and, since the 1990s, there have been reports on click recognition rates in relation to dural puncture resistances produced by different spinal needles [5-7]. Existence of the click has been recognized and is referred to as the “dural puncture click”, but detailed studies clarifying from which tissue the click originates have not been performed. Since Whitacre’s report, spinal needles have undergone structural improvements, and fine-gauge pencil-point needles (25G or thinner) were developed and are currently recommended to reduce the risk of post-spinal headache (PSH) [8-11]. However, due to their thin structure, length, and bending during passage through skin, subcutaneous tissues, muscles, and interspinous and flavum ligaments before reaching the dura mater, they have been more difficult to use. To overcome these technical difficulties and prevent tissue particles from being carried into the subarachnoid space (“coring”) [12], spinal needle introducers have been developed (double-needle technique) [13]. In addition, it has been observed that click recognition is less apparent with fine-gauge, than with large-gauge needles [4,5,14], and the incremental needle advancement technique has been recommended [1].” 

Page 4, Lines 85–92 have been also corrected to clarify the study rationale:

“We hypothesized that fine-gauge spinal needles would produce subjectively perceivable click sensations upon puncture of synthetic dura or fresh porcine dura mater, and that these click sensations would be distinguishable among various spinal needle types, and would reflect objectively measured forces elicited by punctures of either synthetic dura and fresh porcine dura mater. If our hypothesis was true, it would prove that the click sensation is produced, at least in part, by puncture of the isolated dura mater. Furthermore, we aimed to examine whether the artificial dura demonstrates similar characteristics to fresh porcine dura mater.”

Materials and Methods section

The additional information on introducers has been added: “…all needles were advanced through introducers included in the needle kits, and were used with stylets in place.” (Page 5, Lines 113–114 in the revised manuscript)

Discussion 

Two paragraphs have been added at the beginning of the section (Page 16-17, lines 355-382 in the corrected manuscript) discussing the rationale for our hypothesis, as follows:

“Click sensation, previously recognized by Whitacre [4] and others [5-7], has been commonly associated with puncture of the dura mater and, accordingly, referred to as the “dural puncture click” or “loss of resistance” sensation upon puncture of the dura [1-3]. This observation was plausible since the click was usually experienced at the final stages of needle progression just before confirming the cerebrospinal fluid reflux, and it was also sensed with sleeved spinal needles advanced through heel lumens of Tuohy needles used for combined spinal epidural anesthesia [20]. Though associated with the dura mater puncture, the click might have originated from other tissues penetrated by the spinal needle (e.g., skin, subcutaneous tissues, muscles, interspinous and flavum ligaments) before reaching the dura. Thus, the “loss of resistance” sensation felt after puncture of the ligamentum flavum might have been recognized as the click [12]. However, this “loss of resistance” observed with epidural needles has not been uniformly experienced or referred to as a “click” per se, even though Tuohy needles have much larger diameters. The fact that the “click” has not been commonly recognized with Tuohy needles might be related to differences in the elasticity of tissues they penetrate as compared to that of the dura mater.

The prevailing consensus that a puncture of the dura mater produces the click sensation has been therefore not based on evidence. In this regard, 1) the click has been experienced as a subjective sensation during puncture of all tissue layers penetrated before the needle reaches the subarachnoid space; 2) objective measurements of forces produced solely by puncture of the dura mater have not been performed; and 3) the exact clinical settings with patients receiving spinal anesthesia have been impossible to reproduce experimentally. We hypothesized that if puncture of the dura mater alone produces the click sensation, the click may be legitimately referred to as “dural puncture click”, even if it only partially contributes to the overall click sensation in the clinical settings. We thought that this could only be examined with an isolated dura mater (fresh porcine dura mater in our study) and with experimental settings similar to those found in clinical situations when the influence of other tissues has been eliminated; that is, when fine-gauge spinal needles are advanced through needle introducers.”

A paragraph on introducers has been also added to the Limitations (Page 23, Lines 522–528), as follows:

“(a) In clinical settings, the introducer tip does not necessarily reach the epidural space and in such circumstances, the spinal needle must penetrate the remaining tissues (interspinous and flavum ligaments) before puncturing the dura. This might influence the magnitude of click sensation; in our experimental settings this influence was not analyzed. Thus, our experimental settings would have resembled those with the ideal position of the introducer needle in the epidural space or those of combined spinal epidural anesthesia performed with the sleeved needle-in-needle technique.”

We thought that at the present stage of analysis, the Reviewer’s suggestion of adding other tissues (or their models) into the analysis is theoretically valid but too elaborate to be practically and fully addressed. This is due to the following reasons:

1) It has not been defined and clarified how to analyze such tissues. Should their influence on the click sensation and relation to needle characteristics be studied separately for each tissue layer or for all tissues together? Notwithstanding the fact that any of these options would be difficult to implement in experiments with patients (e.g., intra- and inter-participant differences, influence of age, physical status, anatomical differences) due to ethical reasons, the effect of possible changes in elasticity and water content in cadaver dura mater and tissues above on the click sensation (both subjectively felt and objectively measured as force) would have to be addressed if cadavers were to be used.

2) There is no reliable artificial model of all the punctured tissue layers available. Spinal anesthesia simulators have been developed but their purpose is mainly educational and focused on technical aspects of the procedure.

3) Animals or animal tissues might be used in experiments but differences in the characteristics of these tissues when compared to human tissues would have to be addressed in both study designs and result analyses.

With the above-mentioned concerns in mind, we selected the fine-gauge spinal needles that require introducers for spinal puncture experiments. The introducers completely eliminated the influence of tissues above the dura mater (in clinical settings it does not always have to be true as stated in the Limitations section), and therefore, the effects of the isolated dura mater (or its substitute, the PA film) on the click sensation could be examined. The magnitude of the click, both subjectively felt and objectively measured as force difference upon puncture of the dura mater, could be assessed; these findings may serve as a reference for further studies, which are certainly necessary. In our opinion, our experimental settings limited only to fine-gauge needles with introducers could have been closer to the respective clinical ones (also in regard to number of factors influencing the click sensation and forces) than those in which all tissues (or their models) were incorporated into experimental models.

---

## [Editor Report · Decision Letter 1]

7 Jul 2021

Recognition and Differentiation of Dural Puncture Click Sensation:

A Subjective and Objective Prospective Study of Dural Puncture Forces Using Fine-Gauge Spinal Needles

PONE-D-21-03864R1

Dear Dr. Utsumi,

We’re pleased to inform you that your manuscript has been judged scientifically suitable for publication and will be formally accepted for publication once it meets all outstanding technical requirements.

Kind regards,

Jonathan H Sherman

Academic Editor

PLOS ONE
---

## [Editor Report · Acceptance letter]

22 Jul 2021

PONE-D-21-03864R1 

Recognition and Differentiation of Dural Puncture Click Sensation: A Subjective and Objective Prospective Study of Dural Puncture Forces Using Fine-Gauge Spinal Needles 

Dear Dr. Utsumi:

I'm pleased to inform you that your manuscript has been deemed suitable for publication in PLOS ONE. Congratulations! Your manuscript is now with our production department. 

Kind regards, 

on behalf of

Dr. Jonathan H Sherman 

Academic Editor

PLOS ONE